# The BICD2 dynein cargo adaptor binds to the HPV16 L2 capsid protein and promotes HPV infection

Kaitlyn Speckhart[1,2☯], Jeongjoon Choi[3☯], Daniel DiMaio[3], Billy Tsai[1,2]*

**1** Department of Cell and Developmental Biology, University of Michigan Medical School, Ann Arbor, Michigan, United States of America, **2** Cellular and Molecular Biology Program, University of Michigan Medical School, Ann Arbor, Michigan, United States of America, **3** Department of Genetics, Yale School of Medicine, New Haven, Connecticut, United States of America

☯ These authors contributed equally to this work.
* btsai@umich.edu

**Data Availability Statement:** All relevant data are within the manuscript and its Supporting information files.

**Funding:** The study is supported by the National Institutes of Health R01AI150897 (to B.T. and D.

## Abstract

During entry, human papillomavirus (HPV) traffics from the endosome to the *trans* Golgi network (TGN) and Golgi and then the nucleus to cause infection. Although dynein is thought to play a role in HPV infection, how this host motor recruits the virus to support infection and which entry step(s) requires dynein are unclear. Here we show that the dynein cargo adaptor BICD2 binds to the HPV L2 capsid protein during entry, recruiting HPV to dynein for transport of the virus along the endosome-TGN/Golgi axis to promote infection. In the absence of BICD2 function, HPV accumulates in the endosome and TGN and infection is inhibited. Cell-based and *in vitro* binding studies identified a short segment near the C-terminus of L2 that can directly interact with BICD2. Our results reveal the molecular basis by which the dynein motor captures HPV to promote infection and identify this virus as a novel cargo of the BICD2 dynein adaptor.

## Author summary

HPV traffics from the endosome, the TGN/Golgi, and then the nucleus to cause infection, although how this is accomplished is unclear. In this manuscript, we demonstrate that the dynein cargo adaptor BICD2 interacts directly with the HPV L2 capsid protein, ferrying the virus from the endosome to the TGN/Golgi in order to promote infection.

## Introduction

Human papillomavirus (HPV), a non-enveloped DNA virus, causes >95% of cervical cancer, as well as cancer of other tissues including the oropharynx [1]. HPV-associated cancers remain a public health burden despite the availability of a prophylactic vaccine. Additionally, there are currently no antiviral treatments to target established HPV infections. Therefore, it remains

D.), R35CA242462 (to D.D.), and F31AI169887 (to K.S.). The funders had no role in study design, data collection and analysis, decision to publish, or preparation of the manuscript.

critical to gain a more complete understanding of HPV cellular entry as these new insights may reveal new therapeutic strategies.

HPV is composed of the major capsid protein L1 and the minor capsid protein L2. Together they encapsulate the ~8-kb double-stranded DNA genome [2]. HPV infection begins when L1 binds to heparin sulfate proteoglycans on the plasma membrane of basal keratinocytes [3–7]. Subsequent conformational changes of L1 allow for furin cleavage of L2 and the transfer of the viral particle to an unknown entry receptor [7–10]. Upon endocytosis and delivery to the endosome, the viral capsid partially disassembles [11,12]. The host component p120 catenin then targets the virus to the transmembrane protein γ-secretase [13], which together with the action of cell-penetrating peptide (CPP) located at the C-terminus of L2 promotes insertion of the C-terminal segment of the L2 protein across the endosomal membrane [14–16]. This critical step in infection exposes much of L2 to the cytosol, allowing the virus to recruit host factors including Rab9a, Rab7, and retromer that guide the virus from the endosome to the trans-Golgi network (TGN)/Golgi apparatus [17–23]. In the next step of HPV entry, the TGN/Golgi-localized COPI sorting complex captures HPV via interaction with the cytosol-exposed region of L2, ferrying the virus through the Golgi stack *en route* to the nucleus where replication occurs [24].

To reach the nucleus from the cell surface, HPV must navigate the intracellular endomembranous system of the host cell. Previous studies suggest that the virus hijacks the microtubule network via the cytoplasmic dynein motor during infection in order to navigate this complicated membranous system [25–29]. Structurally, dynein is composed of multiple subunits, including the heavy chain, the light chain, and the intermediate chain [30]. Chemical inhibition of the dynein ATPase activity or loss of the dynein light chain prevents HPV from reaching the nucleus to cause infection [26–28]. Specifically, dynein light chain 3 was shown to be required for nuclear import of HPV and association of L2 with mitotic chromosomes [27]. Despite these findings, it is not known whether dynein plays a role in earlier steps of HPV entry and, if so, how dynein is recruited to HPV during these earlier steps.

Dynein is normally linked to a cellular cargo via a dynein cargo adaptor, which provides cargo specificity for the motor [31,32]. We therefore hypothesized that dynein-mediated transport of HPV during entry requires a select dynein cargo adaptor. Here, utilizing proteomics, gene knockdown (KD), and imaging approaches in concert with cell-based and *in vitro* binding studies, we identified the cargo adaptor BICD2 as critical for the endosome-TGN/Golgi trafficking of HPV by recruiting HPV to dynein. BICD2 interacts with HPV throughout this pathway, and this interaction is direct. Additionally, we found a 20-amino acid segment near the C-terminus of L2 is sufficient to bind to BICD2 *in vitro*. Together, these results clarify the mechanism by which dynein recruits HPV to promote infection and reveal the incoming viral particle as a novel cargo for BICD2.

## Results

### HPV16 binds to dynein and the cargo adaptor BICD2

To study HPV cellular entry, we used a well-established pseudovirus (PsV) system consisting of the viral capsid proteins L1 and L2 [33, 34]. In place of the viral genome, a reporter plasmid expressing green fluorescent protein (GFP), HcRed, or luciferase is packaged in the PsV particle. The L2 C-terminus has a 3x-FLAG epitope appended to it (this L2 protein is designated L2F; HPV16 PsV containing L2F is designated HPV16.L2F), allowing for antibody-based detection of L2 throughout the infectious pathway [35]. The key trafficking events of HPV PsV mirror that of HPV generated in stratified raft cultures [33]. Thus, the use of HPV PsV to study viral entry provides insights to native HPV infection events.

We previously used a cell fractionation-immunoprecipitation-mass spectrometry (IP-MS) strategy to identify host factors that bind to Golgi-localized HPV16.L2F PsV [24]. In this approach, Golgi fractions were isolated from HeLa cells infected with HPV16.L2F for 22 hours. HPV16.L2F was immunoprecipitated (IPed) from the Golgi fractions, and the precipitated material was analyzed by MS. As a negative control, Golgi fractions from an uninfected HeLa cell were mixed with purified HPV16.L2F and the sample was subjected to the same IP-MS protocol; we consider any host factor identified under this condition to represent cellular factors that bound to L2 that are not involved in the virus entry pathway. Importantly, we identified peptides corresponding to the heavy chain of the host motor protein dynein in samples derived from infected but not from uninfected cells (Fig 1A). These results suggest dynein is a host factor that interacts with HPV during cell entry, consistent with previous studies where the dynein light chain was identified as an HPV-interacting protein [25,27,28].

To confirm the IP-MS result and to determine when the HPV-dynein interaction occurs during entry, we performed a time course co-IP experiment. HeLa cells were infected with HPV16.L2F for various times, and the resulting cell extracts were subjected to immunoprecipitation using an antibody against dynein intermediate chain (IC). The precipitated material was analyzed by SDS-PAGE and immunoblotting using a FLAG antibody to detect L2F. Our results showed that IP of dynein IC pulled down L2F by 8 hpi and the interaction continued through 24 hpi, whereas a control antibody did not pull down L2F (Fig 1B, top). Previous studies established that HPV16 reaches the endosome by 8 hpi, TGN/Golgi apparatus by 16 hpi, and the nucleus by 24 hpi [14,17,18,26,27,29,35,36]. Hence, these results suggest that dynein interacts with HPV throughout the infectious pathway, ferrying the virus along the endosome-TGN/Golgi-nucleus axis.

Because dynein typically uses select cargo adaptors to interact with cellular cargoes, we hypothesized that a dynein cargo adaptor must physically link dynein to L2. Our IP-MS experiments identified host factors that bound to HPV16 in the Golgi fraction. Therefore, we asked if the Golgi-concentrated dynein cargo adaptor BICD2 binds to HPV16. To test this, we performed a similar time course co-IP experiment as in Fig 1B. Extracts from infected HeLa cells were subjected to IP using an anti-BICD2 antibody. As was the case for dynein IC IP, IP of BICD2 co-precipitated L2F by 8 hpi through 24 hpi (Fig 1C, top panel), while a control antibody did not pull down L2F (Fig 1D). These are the same time points in which the dynein-L2 interaction occurs, suggesting that HPV16 interacts with both dynein and BICD2 throughout the virus infectious pathway.

To obtain further evidence that the HPV16-BICD2 interaction occurred during entry, we used the proximity ligation assay (PLA) in which proteins of interest are targeted with antibodies that will generate a fluorescent signal if the proteins are within 40 nm of each other. HeLa cells were infected with HPV16.L2F for 16 hours and subjected to PLA to determine whether L1 and BICD2 are in proximity during entry. As a control, infected HeLa cells were treated with the chemical γ-secretase inhibitor XXI which blocks endosomal membrane insertion of L2 and exposure of L2 to the cytosol, events required for transport of HPV to the Golgi [14,35,37,38]. HPV16.L2F-infected cells showed significant L1-BICD2 PLA signal compared to mock infected cells, and this signal was markedly reduced in infected cells treated with XXI (Fig 1E; quantified in 1F). These results in intact cells are consistent with the co-IP findings, supporting the idea that the HPV associates with BICD2 during virus entry.

## A short segment near the L2 C-terminus binds directly to BICD2

To determine if the BICD2-L2 interaction is direct, we purified FLAG-BICD2, EGFP-FLAG (a control protein) and HA-(Δ1–67) L2F (Fig 2A). The HA-(Δ1–67) L2F fragment contains the

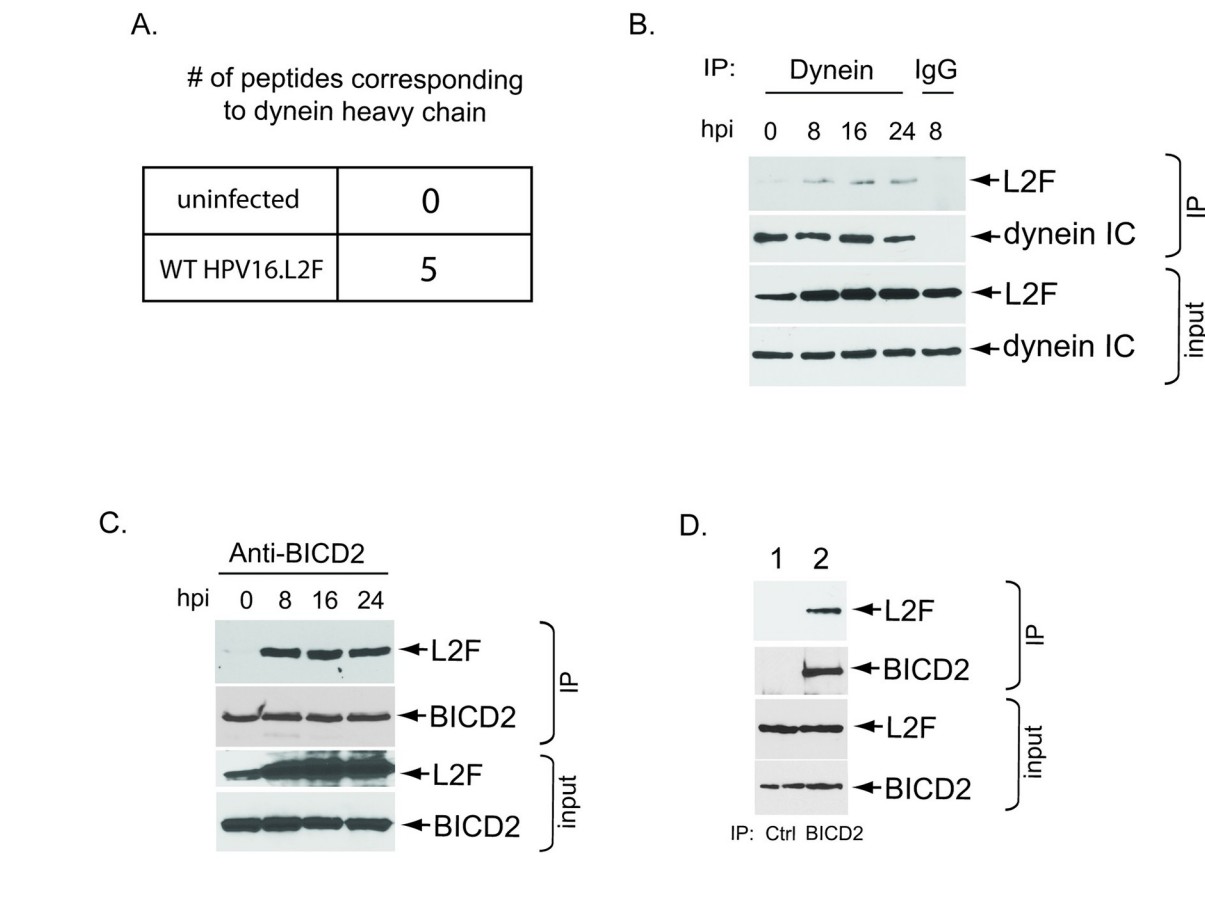

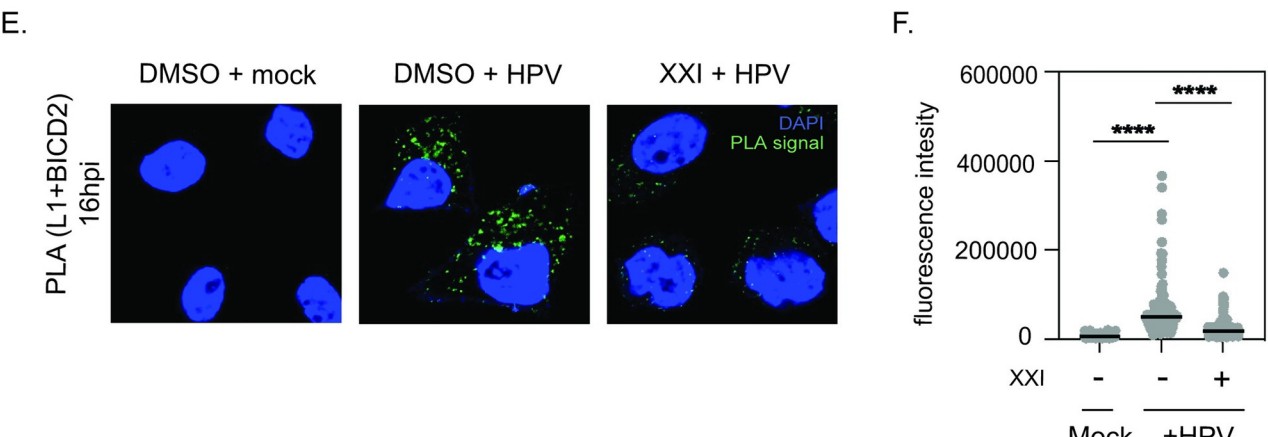

**Fig 1. HPV16 binds to dynein and the cargo adaptor BICD2. A.** Number of different peptides corresponding to dynein heavy chain that co-IPed with L2F identified by mass-spectrometry performed using Golgi fractions derived from either uninfected HeLa cells incubated with purified HPV16.L2F or HPV16.L2F-infected HeLa cells as described in Harwood et al., 2023. **B.** Whole cell extracts derived from HeLa cells infected with HPV16.L2F for the indicated hours post infection (hpi) were subjected to immunoprecipitation with an anti-dynein intermediate chain (IC) antibody or mouse IgG as a negative control. The precipitated material was analyzed by SDS-PAGE and immunoblotting using anti-FLAG and anti-dynein IC antibodies. 0 hpi denotes cells infected for 5 minutes. In all samples, extracellular virus was removed by washing cells with 300mM NaCl before harvesting. **C.** As in (B), except an anti-BICD2 antibody was used for immunoprecipitation. **D.** Whole cell extracts derived from HPV16.L2F-infected HeLa cells (for 8 hours) were subjected to immunoprecipitation as in (C) or with a rabbit IgG antibody as a negative control. **E.** HeLa S3 cells were infected at the multiplicity of infection (MOI) of ~200 with HPV16.L2F containing the HcRed reporter plasmid. Dimethylsulfoxide (DMSO) or γ-secretase inhibitor dissolved in DMSO (XXI) were added to the medium 30 min prior to infection. At 16 hpi, PLA was performed with antibodies recognizing HPV L1 and BICD2.

Mock, uninfected. PLA signals are green; nuclei are blue (DAPI). Similar results were obtained in two independent experiments. **F.** The fluorescence of PLA signals was determined from multiple images obtained as in (E). Each dot represents an individual cell (n>40) and black horizontal lines indicate the mean value of the analyzed population in each group. ****p< 0.0001. The graph shows results of a representative experiment.

portion of L2 thought to be exposed to the cytosol (where BICD2 is located) when L2 is inserted into the endosome membrane [19]. When FLAG-BICD2 was incubated with either EGFP-FLAG or HA-(Δ1–67) L2F and subjected to IP with a BICD2 antibody, HA-(Δ1–67) L2F but not EGFP-FLAG was pulled down (Fig 2B, top panel, compare lane 4 to 3). HA-(Δ1–67) L2F incubated in the absence of BICD2 was not pulled down with the BICD2 antibody (Fig 2B, lane 2), demonstrating that BICD2 is required to co-precipitate HA-(Δ1–67) L2F. These results demonstrate that BICD2 can bind directly to the portion of L2 that is exposed in the cytosol during entry.

To identify a sequence motif within L2 responsible for BICD2 binding, we conducted pull-down experiments with biotinylated peptides derived from three regions of L2 (Fig 2C). Peptide N is composed of amino acids 12 through 44 located at the N-terminus of L2 and is not exposed to the cytosol when L2 is membrane-inserted [19]. Peptide M is composed of amino acids 299 through 312, whose sequence has been shown to be critical for infection because it binds to COPI [24]. Additionally, we probed a region close to the L2 C-terminus by using peptide C consisting of amino acids 434 through 461, which contains the retromer binding site and CPP.

We tested the ability of each peptide to bind to BICD2. Purified FLAG-BICD2 (Fig 2A) was incubated with each of these peptides or the control 3xFLAG peptide. Streptavidin beads were used to pull down the peptide and the bound samples were analyzed by SDS-PAGE and immunoblotting with anti-FLAG to detect BICD2. Strikingly, FLAG-BICD2 precipitated with the L2 C peptide and not the control or the other L2 peptides (Fig 2D), suggesting that BICD2 binds specifically to this 28-amino acid segment of HPV16 L2 (amino acid 434–461). Because this peptide encompasses the retromer binding site and CPP (Fig 2C, highlighted in green and blue, respectively) [16,17], we tested if those elements are important for BICD2 binding. Substitution of either the retromer binding site or the CPP with alanines (to generate CDM and C6A, respectively) severely impaired BICD2 binding (Fig 2E), indicating that the BICD2 binding site overlaps with the retromer binding site and the CPP.

We next used a shorter 20-amino acid peptide consisting of amino acids 442 through 461 (Fig 2C, peptide C442) to further pinpoint the BICD2 binding region. This peptide was incubated with either purified FLAG-BICD2 or purified EGFP-FLAG. After streptavidin pulldown and immunoblotting, we found that the L2 442–461 peptide pulled down FLAG-BICD2 but not the control EGFP-FLAG (Fig 2F, compare lane 4 to 3). Importantly, peptides C442 and C pulled down a similar level of FLAG-BICD2 (Fig 2G, compare lane 5 to 3), indicating the 20 amino acids in the peptide C442 is sufficient for BICD2 binding. To test whether the sequence downstream of the CPP affects L2 binding with BICD2, we used peptides Cx and C442x, which consist of peptides C and C442 containing the amino acids downstream of CPP to the C-terminus of the native L2 protein (Fig 2C). The presence of these amino acids did not affect the amount of BICD2 pulled down (Fig 2G).

To ensure these results were not artifacts of using purified proteins, we performed similar experiments using cell extracts. We incubated indicated peptides with a whole cell extract containing endogenous BICD2 or an extract from cells expressing either the control protein HA-mCherry or HA-BICD2-mCherry. After streptavidin pulldown and immunoblotting, we found that the peptide C precipitated endogenous BICD2 while the peptide M did not, consistent with the data using purified FLAG-BICD2 protein (S1A Fig). Additionally, both peptides

 

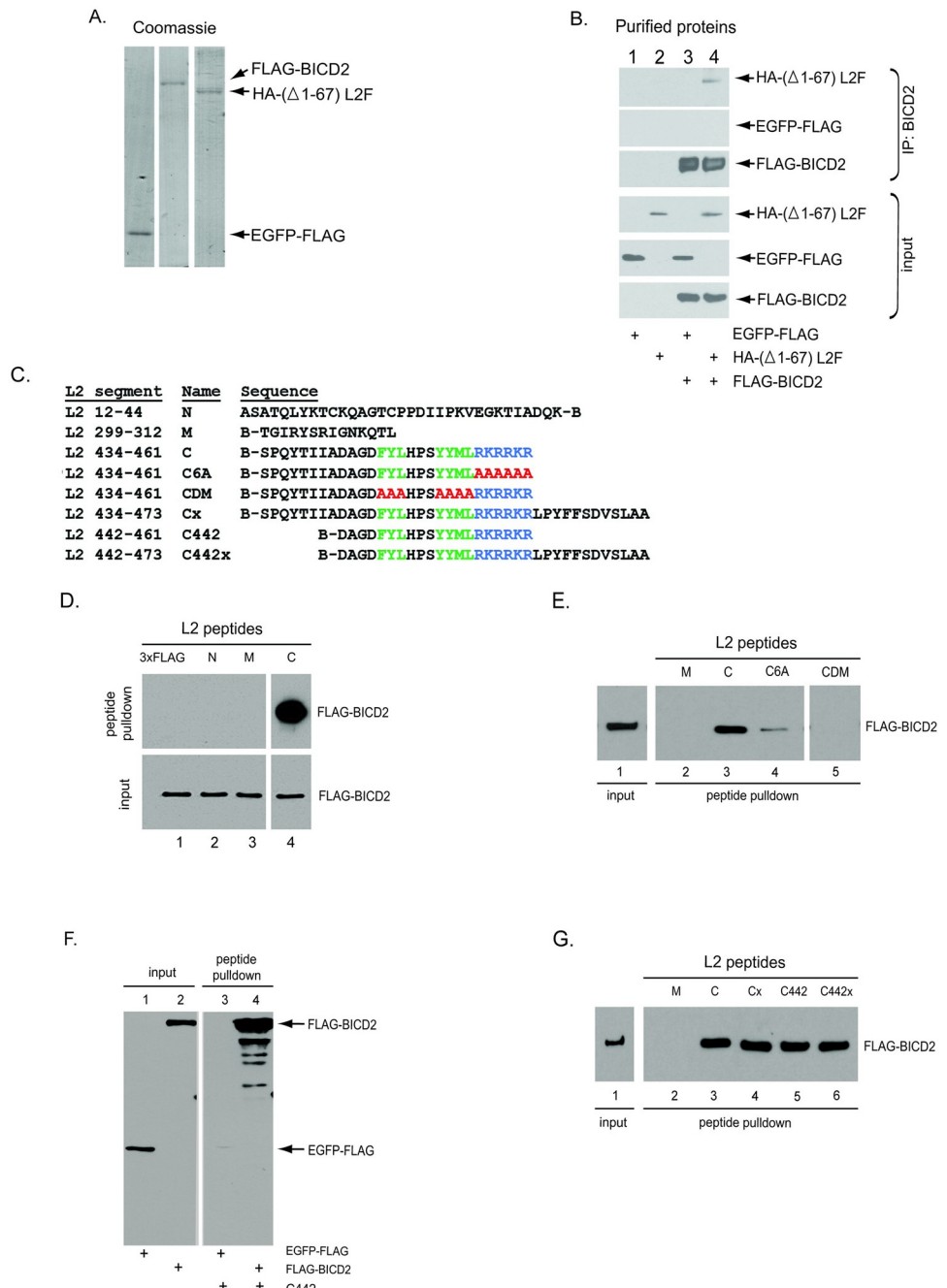

**Fig 2. A short segment near the L2 C-terminus binds directly to BICD2. A.** Coomassie stain of SDS-PAGE. Left to right: purified EGFP-FLAG, FLAG-BICD2, and HA-(Δ1–67) L2F. **B.** FLAG-BICD2 was incubated with either EGFP-FLAG or HA-(Δ1–67) L2F. Immunoprecipitation was performed with an anti-BICD2 antibody, and the precipitated material was analyzed by SDS-PAGE and immunoblotting with an anti-FLAG antibody. **C.** Sequences of the HPV16 L2 peptides. B indicates biotin. Amino acids of the retromer binding site are shown in green, those of the CPP are shown in blue, and mutated amino acids are shown in red. **D,E.** FLAG-BICD2 was incubated with the indicated peptide; biotinylated L2 peptide or 3xFLAG peptide as a control. Precipitation was performed using streptavidin beads. The precipitated material was subjected to SDS-PAGE and immunoblotted with an antibody recognizing FLAG. All samples were electrophoresed on the same gel. Irrelevant lanes were removed. **F.** A shorter biotinylated peptide (C442) was incubated with FLAG-BICD2 or with EGFP-FLAG as a control and analyzed as described in (D). **G.** As in (D) and (E).

C and C442 pulled down HA-BICD2-mCherry, but not the HA-mCherry control (S1B Fig, lanes 3–5). Taken together, these results show that this 20-amino acid C-terminal segment of HPV16 L2 is sufficient for binding to BICD2.

Because this segment of L2 is important for the binding of multiple proteins, we further characterized how retromer and BICD2 interact with this segment, once again using purified proteins. The retromer complex (comprising VPS26, VPS29, and VPS35) was purified by using VPS29-FLAG (S1C Fig). Wild-type peptide C pulled down retromer and the CDM mutant lacking the retromer binding site was defective, as expected (S1D Fig). The CPP mutant peptide (designated C6A) was also defective for retromer binding. When incubated with both retromer and BICD2, the peptide C pulled down both proteins (S1E Fig, lane 4). These data suggest either that this segment of L2 can bind retromer and BICD2 simultaneously or that some peptides in the reaction bind solely to retromer while others bind BICD2.

## BICD2 promotes infection of multiple HPV types

As BICD2 acts as a dynein adaptor and dynein is necessary for HPV entry, we then asked if the dynein cargo adaptor BICD2 plays a role in virus infection. Within the BICD family of cargo adaptors are BICD1, BICD2, and the more distantly related BICDR1 and BICDR2 [39,40]. To test if BICD2 and the other BICD family members promote HPV16 infection, we used siRNAs to knockdown (KD) each of these proteins. Two individual siRNAs targeting each of these adaptor proteins efficiently reduced the intended protein levels in HeLa cells (S2A Fig). Cells were infected with HPV16.L2F containing the GFP reporter plasmid. Flow cytometry was used to analyze infection, with the expression of GFP used as an indication of successful viral nuclear arrival. Depletion of any of the adaptors decreased HPV16.L2F infection to a varying extent when compared to cells treated with the scrambled (Scram) siRNA, with KD of BICD2 and BICDR1 blocking infection most severely (Fig 3A). Given these results, we performed peptide pulldown using HeLa cell lysates to determine if peptide C could bind BICDR1. Similarly to S1A Fig, we found that peptide C bound endogenous BICDR1 (S1F Fig), further suggesting that HPV16 uses two dynein adaptors during infection. Membrane integrity under all KD conditions was largely unaffected compared to control cells (S2B Fig). Because loss of BICD2 or BICDR1 resulted in the most severe phenotype in HeLa cells, we also knocked down each of these proteins in HaCaT cells (S2C Fig) and found that infection was also impaired in these cells (Fig 3B) without damaging the cellular membrane (S2D Fig).

We then evaluated the importance of BICD2 and BICDR1 in promoting infection of different HPV types. KD of BICD2 (but not BICDR1) inhibited infection of HPV5 (Fig 3C) and HPV18 (Fig 3D), suggesting that BICD2 is critical for infection of multiple HPV types. Therefore, we focused the rest of our study on BICD2. The C-terminus of HPV5 and HPV18 L2 corresponding to HPV16 L2 434–461 are conserved, including the retromer binding site and CPP (S3A Fig). Peptides corresponding to HPV16 L2 434–461 for HPV5 (5C) and HPV18 (18C) also bound purified BICD2 (S3B and S3C Fig).

We further assessed the effect of BICD2 knockdown on other aspects of cell behavior. Golgi morphology as assessed by cell staining using the Golgi marker GM130 was not affected by BICD2 KD (S2E Fig). Furthermore, co-localization of the TGN marker TGN46 with endosomal marker EEA1 was not altered by BICD2 KD (S2F Fig). Additionally, cell cycle progression (as assessed by analyzing the percent of cells in $G_1$, S, or $G_2$-M phases) was also largely unaffected by BICD2 KD (S2G and S2H Fig). Together, these data demonstrate that loss of BICD2 inhibits infection by several HPV types but does not globally compromise cellular integrity, suggesting BICD2 serves a specific function during HPV infection.

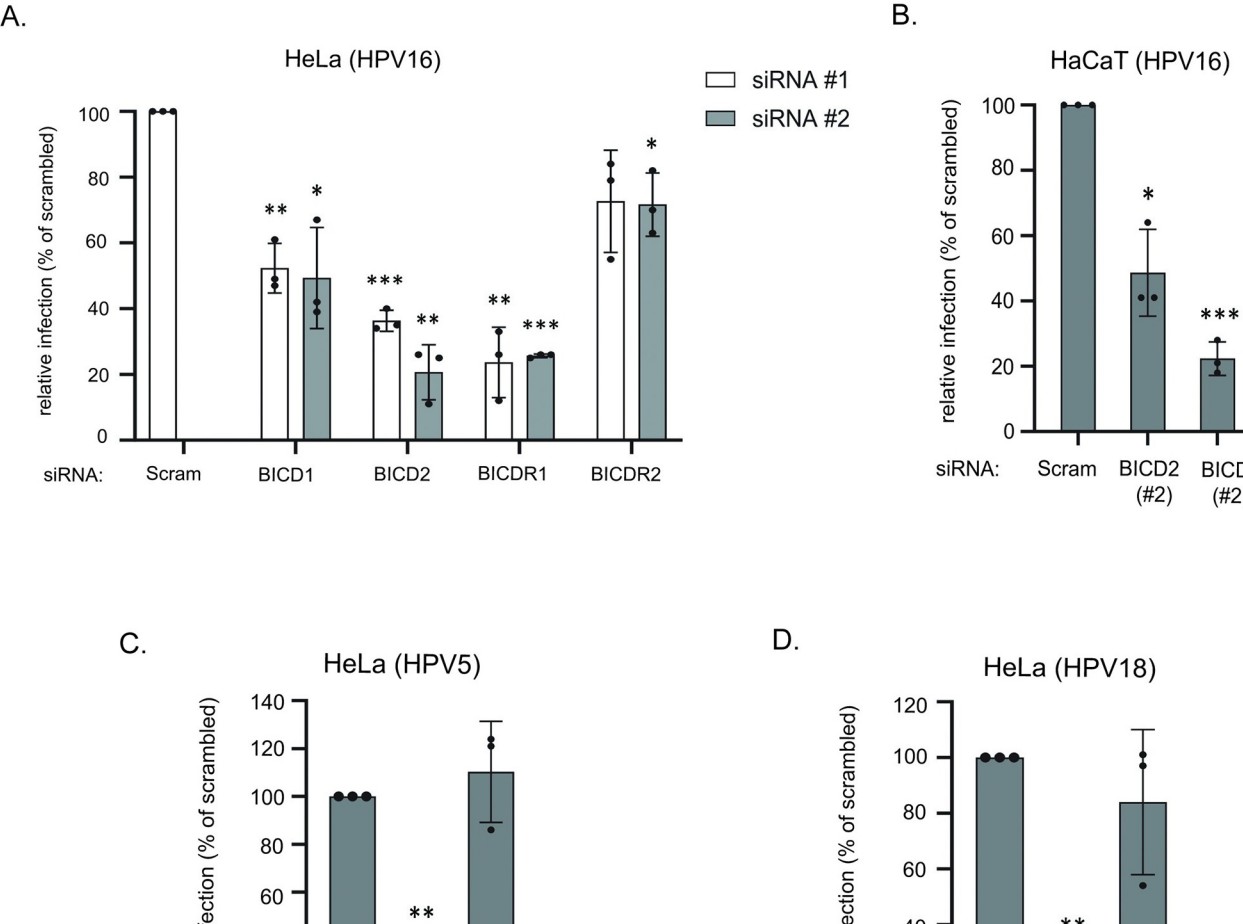

**Fig 3. BICD2 promotes infection of multiple HPV types. A.** HeLa cells transfected with the indicated siRNA were infected with HPV16.L2F containing a GFP reporter plasmid. Flow cytometry at 48 hpi was used to determine the fraction of cells that were GFP-positive. The data was normalized to infected cells transfected with scrambled siRNA. The means and standard deviations (SD) are shown of at least three independent experiments. **B.** As in (A), except human HaCaT keratinocytes were used. **C, D.** As in (A), except cells were infected with HPV5 (C) or HPV18 (D) pseudovirus containing a reporter plasmid expressing GFP and luciferase separately, and a single siRNA targeting BICD2 or BICDR1 was used. Infection was measured by GFP expression. In all panels, indicated p values are compared to the scrambled condition. *p ≤ 0.05, **p ≤ 0.01, ***p ≤ 0.001.

## BICD2 acts as a cargo adaptor linking dynein to HPV L2 during entry

BICD2 is known to act as an adaptor that links cargo to dynein [41,42]. To determine whether BICD2 acts as an adaptor to recruit HPV to dynein during entry, we used a reported dynein binding-defective BICD2 mutant [31]. Specifically, alanine residues at positions 43 and 44 within full-length BICD2 play an important role in mediating dynein binding. A BICD2 mutant in which these alanines are changed to valine (A/V BICD2) was shown to not bind to dynein [31]. To verify this in our system, HeLa cells were transfected with an HA-tagged wild-type (WT) BICD2*, HA-tagged A/V BICD2*, or an empty vector control (* indicates a mutant

siRNA-resistant version of BICD2). The resulting cell extracts were subjected to IP with an antibody against dynein IC. Immunoblotting showed that WT but not A/V BICD2 co-precipitated with dynein (Fig 4A, compare lane 2 to 3), confirming that the alanine mutations in BICD2 inhibit association with dynein.

We then performed a KD-rescue experiment to evaluate if the ability of BICD2 to interact with dynein is essential for BICD2 to promote HPV infection. HeLa cells were transfected with plasmids expressing the control HA-mCherry, HA-WT BICD2*, or HA-A/V BICD2*. The cells were co-transfected with scrambled control siRNA or BICD2 siRNA. Cells were infected for 48 hours with HPV16.L2F, fixed, incubated with an anti-HA antibody, and examined by microscopy. GFP fluorescence in cells expressing the HA construct was scored so that infection is assessed in only those cells expressing the control or rescue constructs. In cells expressing HA-mCherry, knockdown of BICD2 blocked infection as expected (Fig 4B, first to second bar). Expression of HA-WT BICD2* in cells transfected with the scrambled control siRNA did not increase infection (Fig 4B, third bar). Importantly, expression of HA-WT BICD2* in BICD2 KD cells largely restored infection (Fig 4B, fourth bar), but expression of HA-A/V BICD2* in BICD2 KD cells did not (Fig 4B, fifth bar). As a control, we show that this inability to restore infection is not because A/V BICD2 is globally misfolded because it can bind to HPV L2 during entry (Fig 4C), suggesting that cargo-binding remains intact for this BICD2 mutant. These data demonstrate that the BICD2 KD phenotype is not due to an off-target siRNA effect and suggest that BICD2 relies on dynein interaction to support HPV infection.

Our data suggested that BICD2 must engage dynein to carry out its function during HPV infection. Because BICD2 can act as a cargo adaptor linking dynein to a cellular cargo, we hypothesize that BICD2 acts as a cargo adaptor to connect dynein to the HPV cargo. To test this, we depleted BICD2 and asked if the dynein-L2 interaction during entry is disrupted. Specifically, cells transfected with scrambled or BICD2 siRNA were infected with HPV16 PsV for 16 h, and the resulting cell extract subjected to IP with an anti-dynein IC antibody. Strikingly, whereas L2F immunoprecipitated with dynein in control cells (as expected), the L2-dynein interaction was lost upon BICD2 KD (Fig 4D, compare lane 1 to lane 2). Furthermore, in intact cells, BICD2 KD significantly decreased the PLA signal between L1 and dynein as compared to control cells at 16 hpi (Fig 4E and 4F), indicating that BICD2 is required for normal HPV16-dynein association. Taken together, these data strongly support the idea that BICD2 functions as a cargo adaptor, coupling dynein to the HPV cargo so that this motor can transport the virus during entry.

## Loss of BICD2 causes accumulation of HPV16 in the endosome and TGN/Golgi

If BICD2 couples HPV to dynein during entry, we predict that BICD2 KD will cause a trafficking defect. To test this, the fate of HPV16 in BICD2-depleted cells was assessed by PLA. As shown in Fig 5A, cells infected with HPV16.L2F under BICD2 KD showed an approximately 8-fold increased PLA signal between L1 and EEA1 (an endosomal marker) as compared to control cells at 16 hpi, when incoming HPV has largely departed the endosome of the control cells for the TGN (Fig 5A, top row; quantified in Fig 5B). At 24 hpi and 32 hpi, the PLA signal between L1 and EEA1 was also higher in KD than in control cells (Fig 5A, middle and bottom rows; quantified in Fig 5B), but the extent of increase was more modest, less than three-fold. These data indicate that BICD2 plays a role in escape of HPV16 from the endosome, consistent with the co-IP data showing that BICD2 binds to HPV16 as early as 8 hpi when the incoming virus is in the endosome (Fig 1). However, the less robust increase in PLA signal at later times implies that the block to endosome exit is not complete.

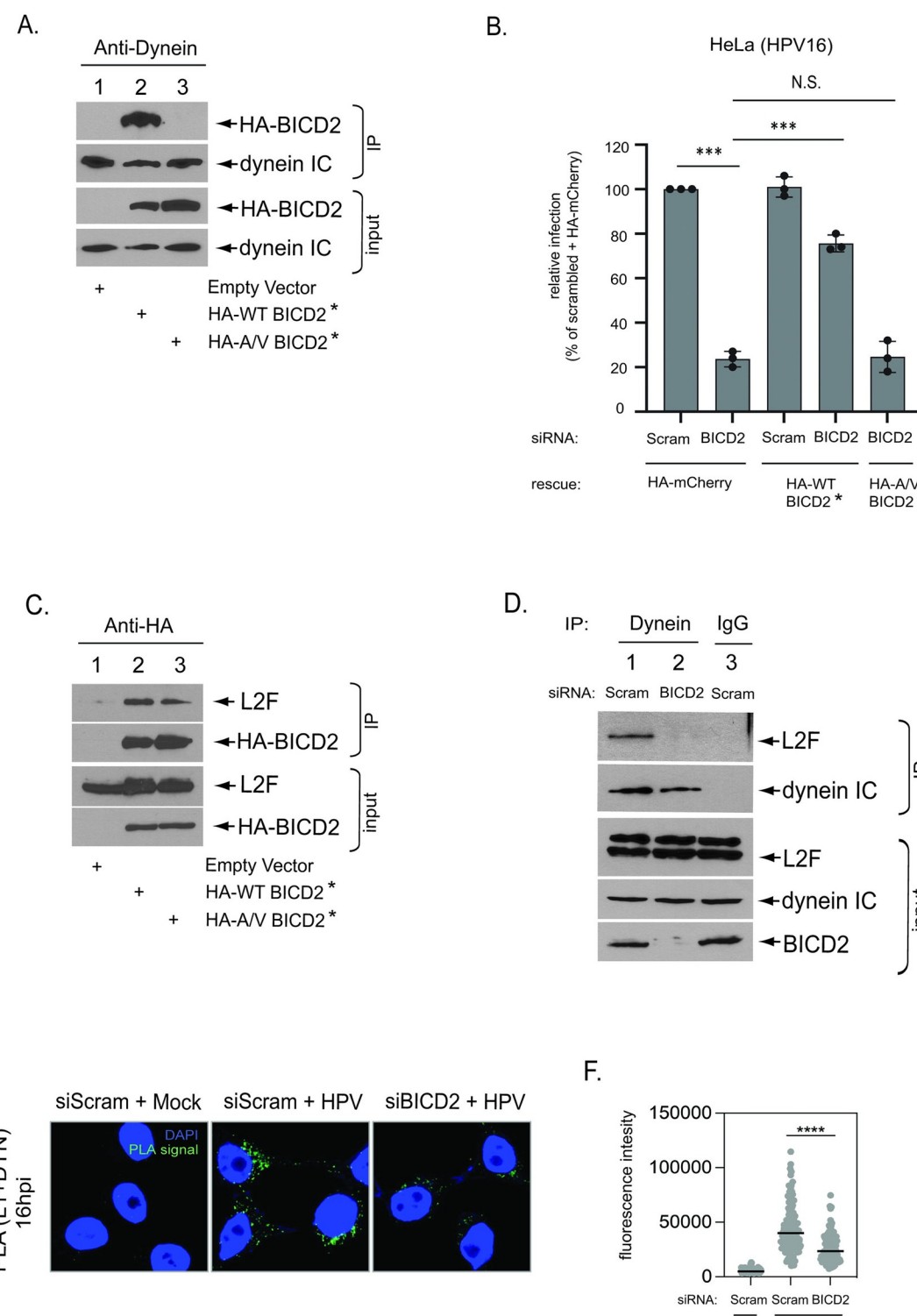

**Fig 4. BICD2 acts as a cargo adaptor linking dynein to HPV L2 during entry. A.** Whole cell extracts derived from HeLa cells transfected with plasmids expressing the indicated protein were subjected to immunoprecipitation with an anti-dynein IC antibody. Precipitated material was analyzed by SDS-PAGE and immunoblotting with anti-HA and anti-dynein IC antibodies. * siRNA resistant. **B.** HeLa cells initially transfected with a plasmid expressing the indicated protein and then transfected with the indicated siRNA were infected with HPV16.L2F containing a GFP reporter plasmid. At 48 hpi, cells were then fixed, permeabilized, subjected to immunofluorescent staining with an anti-HA antibody, and analyzed by fluorescence microscopy. HA-positive cells were

analyzed for GFP fluorescence. The results were normalized to scrambled siRNA-transfected cells expressing HA-mCherry (set at 100%). The means and SD of three independent experiments (where each experiment includes at least counting 100 cells) are shown. Indicated p values are compared to scrambled transfected cells expressing HA-mCherry. ***p ≤ 0.001, N.S., non-significant. **C.** HeLa cells transfected with plasmids expressing the indicated construct were infected with HPV16.L2F for 16 hours and the whole cell extracts subjected to immunoprecipitation with an anti-HA antibody. The precipitated material was subjected to SDS-PAGE and immunoblotting using anti-HA and anti-FLAG antibodies. **D.** HeLa S3 cells transfected with the indicated siRNA were infected with HPV16.L2F. At 16 hpi, whole cell extracts were subjected to immunoprecipitation with an anti-dynein IC antibody. Precipitated material was analyzed by SDS-PAGE and immunoblotting with anti-FLAG, anti-dynein IC, and anti-BICD2 antibodies. **E.** HeLa S3 cells were transfected with scrambled control (siScram) or BICD2-targeting (siBICD2) siRNAs and infected with HPV PsV containing the luciferase reporter plasmid at the MOI of ~200. At 16 hpi, PLA was performed with antibodies recognizing HPV L1 and dynein. Mock, uninfected; HPV, infected. PLA signals are green; nuclei are blue (DAPI). Similar results were obtained in two independent experiments. **F.** The fluorescence of PLA signals was determined from multiple images obtained as in (E). Each dot represents an individual cell (n>40) and black horizontal lines indicate the mean value of the analyzed population in each group. ****p < 0.0001. The graph shows results of a representative experiment. BICD2 siRNA #2 was used in all knockdown experiments.

After exit from the endosome, HPV is transported to the TGN. Therefore, we further asked if HPV16 also accumulates in the TGN under BICD2 KD. Cells were infected with HPV16 PsV and evaluated by PLA using L1 and TGN46 (a marker for the TGN). The PLA signal at 16, 24, and 32 hpi was increased by 6- to 8-fold upon BICD2 KD when compared to control cells, indicating a more persistent accumulation in the TGN than in the endosome as a result of inhibition of exit from the TGN (Fig 5C; quantified in Fig 5D). The PLA results were not due to altered distribution of EEA1 and TGN46 in BICD2 KD cells, because the co-localization of EEA1 and TGN46 was not affected by BICD2 depletion (S2F Fig). Cells infected by HPV16 PsV containing EdU-labelled reporter plasmid DNA also displayed significantly increased co-localization of DNA with TGN46 in BICD2 KD cells compared to control cells at 24 hpi (S4 Fig). These findings demonstrate that BICD2 also promotes escape of HPV16 from the TGN, consistent with the co-IP studies demonstrating BICD2-HPV L2 interaction at 16 hpi when the virus is in the TGN. Of note, the presence of HPV16 in the TGN upon BICD2 KD shows that the virus is not completely trapped in the endosome under this depleted condition. Together, our results reveal that BICD2 executes a critical role at multiple trafficking steps during HPV entry, most prominently after the virus has exited the endosome.

## Discussion

This study elucidates an important step during HPV infection. After internalization, the virus traffics from the endosome to the TGN/Golgi, and finally to the nucleus to cause infection. Previous reports demonstrated a role of dynein during HPV entry, particularly during the late nuclear entry stage, and we previously conducted an IP-MS screen and reported dynein as a binding partner of HPV localized to the Golgi during entry [24–28]. However, it was not determined how the virus is recruited to dynein and whether dynein plays a role in early stages of entry.

Here, by using cell-based co-IP analysis and imaging studies of infected cells, we show that the dynein cargo adaptor BICD2 binds HPV relatively early during virus entry and recruits dynein to HPV to enable virus transport along multiple steps of the infectious pathway. In other systems, BICD2 acts as an adaptor that binds directly to cargo proteins and to the light intermediate chain of dynein to link cargo to dynein for transport [31,32,41–44]. Although BICD2 is highly concentrated in the TGN/Golgi [42], a pool of BICD2 must also localize to the endosome so that it can engage HPV16 at both compartments. Indeed, previous reports used immunofluorescent staining to detect BICD2 not only in the Golgi but also in other locations, possibly including the endosome [41,42,45]. Furthermore, dynein transports vesicles derived

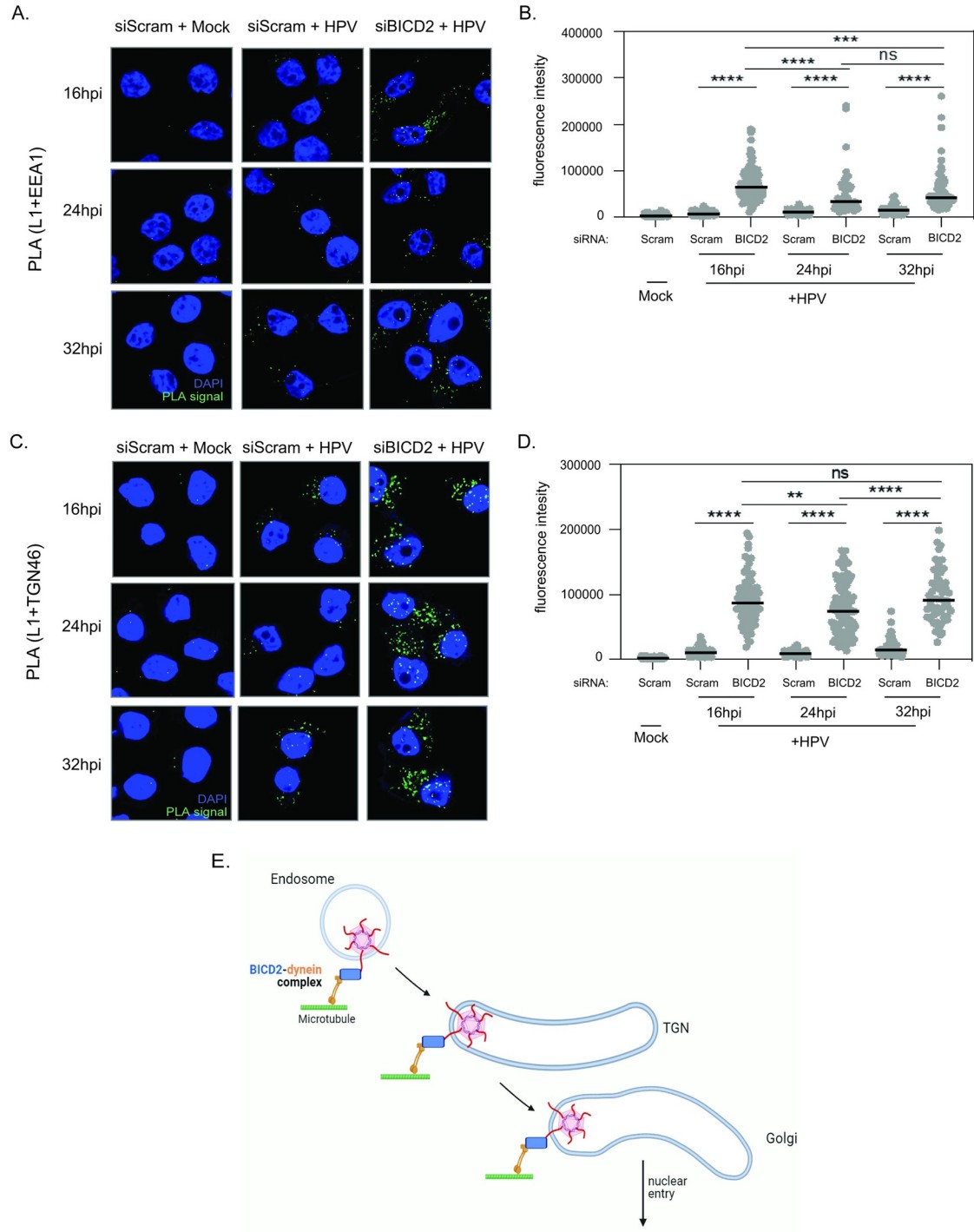

**Fig 5. Loss of BICD2 causes accumulation of HPV in the endosome and the TGN/Golgi. A.** HeLa S3 cells were transfected with scrambled control (siScram) or BICD2-targeting (siBICD2) siRNAs and infected with HPV harboring the luciferase reporter plasmid at the MOI of ~200. At 16, 24, and 32 hpi, PLA was performed with antibodies recognizing HPV L1 and EEA1. Mock, uninfected; HPV, infected. PLA signals are green; nuclei are blue (DAPI). Similar results were obtained in two independent experiments. **B.** The fluorescence of PLA signals was determined from multiple images obtained as in (A). Each dot represents an individual cell (n>40) and black horizontal lines indicate the mean value of the analyzed population in each group. ***p < 0.001; ****p < 0.0001; ns, not significant. The graph shows results of a representative experiment. **C.** As in (A) except PLA was performed using antibodies recognizing HPV L1 and TGN46. **D.** As in (B) from multiple images obtained as in (C). **p < 0.01. **E.** Proposed model for dynein-BICD2 dependent transport of HPV16 during entry. The dynein-BICD2 complex first engages HPV L2 after

cytosolic exposure of L2 from the endosome. This interaction helps to transport the virus from the endosome to the TGN/Golgi. The BICD2-dynein complex continues to ferry HPV through the Golgi and eventually to the nucleus for infection. Figure created in BioRender. Blue, BICD2; orange, dynein; red, L2; green, microtubule; light blue, membrane lipid bilayer. BICD2 siRNA #2 was used.

from both endosome and Golgi [46], supporting the possibility of BICD2-dynein functioning at both endosome and TGN during HPV entry.

We used peptide binding assays with purified proteins and cell extracts to identify a 20-amino acid region near the C-terminus of L2 that is sufficient for BICD2 binding. This result is consistent with published mutational analysis of the full-length L2 protein showing that sequences within the CPP are required for association with dynein in infected cells, although that study also revealed a role for sequences downstream of the CPP [25]. Our data thus support a model in which BICD2 is recruited to HPV once the C-terminal segment of L2 is exposed to the cytosol (Fig 5E). This interpretation is consistent with the finding that treatment with a γ-secretase inhibitor, which blocks L2 protrusion, also inhibited HPV-BICD2 association.

This 20-amino-acid L2 peptide sufficient for BICD2 binding contains two known functional elements, the retromer binding site and the CPP, both of which are required for binding to BICD2 *in vitro*. Because the retromer binding site and the CPP are also necessary during infection for stable exposure of the C-terminus of L2 in the cytoplasm where it can bind BICD2 [16,17,19,39,40], it is difficult to functionally dissect the previously identified roles those elements (i.e., retromer binding and cell-penetrating activity) from the role of BICD2 binding to this segment in infected cells. Thus, the C-terminal segment of L2 is evidently densely packed with binding and functional motifs that may overlap and work in a coordinate fashion to support HPV entry. Furthermore, each HPV capsid contains multiple molecules of L2. Therefore, it is likely that a capsid will bind to both retromer and BICD2 either because a single L2 molecule can bind these two proteins at the same time (or sequentially) or because some L2 molecules bind retromer and other L2 molecules in the same virus particle bind BICD2.

Previous findings using a chemical inhibitor and genetic knockdown studies demonstrated that dynein plays a critical role in HPV16 infection [25–28]. Similarly, our results showed that depletion of the dynein adaptor protein BICD2 blocked HPV infection in multiple cell types, indicating that BICD2 is also crucial during virus infection. Our data suggest that inhibition of HPV infection by BICD2 knockdown is not due to a general defect in intracellular trafficking. The cell cycle is not substantially perturbed by BICD2 knockdown, and postulating a more general trafficking defect does not take into account several of our findings: 1) L2 can bind directly to BICD2, a known dynein adaptor, 2) HPV associates with BICD2 at time points consistent with the accumulation of HPV in the endosome and TGN, and 3) BICD2 KD and BICD2 mutations that prevent dynein interaction inhibit the HPV-dynein interaction and inhibit its ability to support HPV entry. However, because we have not yet identified L2 mutations that inhibit BICD2 binding without affecting other L2 functions (e.g., without inhibiting L2 protrusion), we cannot rigorously test the model that BICD2 binding to L2 is required for proper trafficking.

Using a KD-rescue approach, we found that wild-type BICD2 but not a dynein binding-defective BICD2 mutant can support HPV16 infection, suggesting that the ability of BICD2 to bind dynein is central to its role during HPV entry. Indeed, depletion of BICD2 inhibited the dynein-HPV interaction as measured by coimmunoprecipitation experiments and by PLA in intact cells, demonstrating that BICD2 is required to link dynein to the incoming virus particle. A previous study showed that a large segment of the L2 protein (amino acids 280–473) can

bind to dynein in a yeast 2-hybrid system [28], but our data demonstrated that, in the context of HPV-infected cells, BICD2 is required for L2-dynein association. The ~200 amino acid segment of HPV16 L2 used in the yeast system included the BICD2 binding site mapped here. Thus, it is possible that L2 segments other than the C-terminal 20 amino acids necessary for L2-BICD2 binding may also be involved in L2-dynein binding or that there is a BICD2-like adaptor in yeast mediating the L2-dynein association.

Dynein transports a variety of cargos including organelles, ribonucleoprotein complexes, proteins, and viruses [46,47]. Our study showed that BICD2 is involved in trafficking of HPV through two organelles, the endosome and the TGN. Our time-course analysis revealed that the dynein-BICD2 complex engages HPV16 L2 by 8 hpi when the virus reaches the endosome, where L2 becomes exposed to the cytosol to bind retromer, and the interaction persists during the time the virus transits the Golgi. Furthermore, our finding that BICD2 knockdown causes HPV to accumulate in both the endosome and the TGN indicates that the dynein-BICD2 complex is required for HPV16 to traffic out of the endosome and through TGN/Golgi compartments. We propose that after retromer mediates budding of HPV from the endosome, the dynein-BICD2 complex then transports the virus-containing vesicle to the TGN and then through the Golgi stacks.

Some HPV16 can still reach the TGN/Golgi after BICD2 KD, suggesting that BICD2 KD causes incomplete entrapment of the virus in the endosome. Consistent with this conclusion, HPV16 accumulation in the endosome was substantial at 16 hpi in BICD2 KD cells, but was markedly attenuated at 24 or 32 hpi. In contrast, TGN accumulation was substantial not only at 16 hpi, but also at 24 and 32 hpi. There may be redundant mechanisms to transport HPV from the endosome to the TGN. For example, other dynein adaptors such as BICDR1 may also recruit dynein to HPV16 in the endosome. Alternatively, the incomplete blockade of HPV trafficking from endosome to the TGN may be due to partial BICD2 depletion. Consistent with this latter possibility, our PLA data showed only partial loss of HPV-dynein association upon BICD2 KD. We note that this phenotype caused by BICD2 KD is different from that caused by impaired binding of L2 to retromer, which causes accumulation of HPV16 at the endosome but not at the TGN [17,18,22].

KD of BICDR1 also blocked HPV16 infection, indicating that BICDR1 participates in a critical virus entry step. Because KD of BICD2 but not BICDR1 blocked HPV5 and HPV18 infection, BICD2 appears to be more broadly important than BICDR1 across different HPV types. How HPV16 uses two different dynein adaptors is unclear. One possibility is that while the dynein-BICD2 complex promotes endosome-TGN/Golgi transport of HPV16, the dynein-BICDR1 complex is used to support a different entry step. Alternatively, both BICD2 and BICDR1 may perform the same function, but a "threshold" level of these proteins is necessary to efficiently transport HPV16, and this level is not reached if either adaptor is knocked down. It is also possible that BICDR1 supports HPV infection but not by acting as a dynein cargo adaptor to transport HPV. For example, BICD2 and BICDR1 are required for SV40 disassembly, independently of dynein and dynactin [48].

Additional cellular factors may act in concert with the dynein-BICD2-HPV complex to promote virus transport during entry. Our time-course study indicated that interaction between the dynein-BICD2 complex and L2 persists at times when the virus reaches the nucleus, suggesting that dynein-BICD2 is involved in delivery of HPV16 to the nucleus as well as for exit from the endosome and passage through the TGN/Golgi. In this regard, two nuclear membrane-associated proteins RanBP2 and Nesprin-2 have been shown to interact with BICD2 [49, 50]. It will be interesting to determine whether these host factors are used to recruit the dynein-BICD2-HPV complex to the nuclear membrane as a first step in gaining nuclear entry. In addition, we note that Lai et al [27] proposed that RanBP10/KPNA2 acts as an adaptor for

dynein-mediated trafficking during nuclear HPV entry, raising the possibility that different dynein adaptors may act at different stages during HPV entry, just as different Rab proteins act at different entry stages [22,23]. These proteins may work in an orchestrated series of events, with one "handing off" the virus to the next. Such a hand off may be a result of competition for factors or from an event that triggers one protein to release the virus so that it can be captured by the next protein in the sequence. Alternatively, different host factors could bind separate L2 proteins, as a single viral particle could have multiple L2 proteins exposed to the cytosol.

In sum, we have identified the HPV L2 protein as a novel cargo for the BICD2 dynein adaptor during the early stages of HPV trafficking during virus entry. Beyond HPV, HIV-1 [51] and HSV-1 [52], both enveloped viruses, are the only other viruses reported to use BICD2 for cellular trafficking that leads to nuclear entry. Further analysis of the role of BICD2 and dynein in HPV infection will provide new insights into virus entry and into intracellular protein trafficking in general.

## Materials and Methods

### Antibodies and inhibitors

All antibodies and inhibitors used in this study are listed in Table 1.

**Table 1. Antibodies and inhibitors.**

| Antibody | Species | Source/Catalog no. | Application |
|---|---|---|---|
| Dynein IC 1/2 | Mouse | Santa Cruz; sc-13524 | WB, IP |
| FLAG M2 | Mouse | Millipore Sigma; F3165 | WB |
| BICD2 | Rabbit | Abcam; ab117818 | WB, IP, PLA |
| HA | Rat | Roche; 11867423001 | WB, IF |
| HA | Mouse | Santa Cruz; sc-7392 | IP |
| BICD1 | Rabbit | Millipore Sigma; HPA041309 | WB |
| BICDR1 | Rabbit | Invitrogen; PA5-66367 | WB |
| BICDR2 | Rabbit | Millipore Sigma; HPA043251 | WB |
| GM130 | Rabbit | Abcam; ab52649 | IF |
| Actin | Rabbit | Cell Signaling; 4967S | WB |
| HPV16 L1 | Mouse | BD Biosciences; 554171 | PLA |
| EEA1 | Rabbit | Cell Signaling; 2411 | PLA |
| EEA1 | Mouse | BD Biosciences; 610457 | IF |
| TGN46 | Rabbit | Abcam; ab50595 | PLA |
| Dynein | Rabbit | Invitrogen; PA5-89505 | PLA |
| VPS26 | Rabbit | Abcam; ab23892 | WB |
| Normal mouse IgG | Mouse | Santa Cruz; sc-2025 | WB |
| Normal rabbit IgG | Rabbit | Millipore Sigma; NI01 | WB |
| **Other antibodies** | **Species** | **Source/Catalog no.** | **Application** |
| Anti-mouse IgG peroxidase | Goat | Millipore Sigma; A4416 | WB |
| Anti-rabbit IgG peroxidase | Goat | Millipore Sigma; A4914 | WB |
| Anti-rat IgG peroxidase | Rabbit | Millipore Sigma; A5795 | WB |
| Anti-rat Alexa Fluor 594 | Donkey | Invitrogen; A21209 | IF |
| Anti-rabbit Alexa Fluor 488 | Goat | Invitrogen; A11008 | IF |
| Anti-mouse Alexa Fluor 488 | Donkey | Invitrogen; A21202 | IF |
| Anti-rabbit Alexa Fluor 647 | Goat | Invitrogen; A21245 | IF |
| **Inhibitors** | **Solvent** | **Source/Catalog no.** | **Conc.** |
| XXI | DMSO | Millipore Sigma; 565790 | 2uM |

## Cell culture

HeLa, HeLa S3, and HEK 293T cells were purchased from ATCC. HaCaT cells were purchased from AddexBio Technologies. HEK 293TT cells were a gift from Dr. C. Buck (National Cancer Institute, Rockville, MD). All cells were cultured at 37˚C under 5% $CO_2$ in DMEM (Thermo Fisher Scientific), 10% fetal bovine serum (R&D Systems) and penicillin/streptomycin (Invitrogen).

## DNA constructs

The p16sheLL, p18sheLL, and the p5sheLL plasmids were gifts from Dr. J. Schiller (National Cancer Institute, Rockville, MD; Addgene plasmids #37320, #46953, and #37321) and were used to generate p16sheLL.L2F, p5sheLL.L2F, and p18sheLL.L2F as previously described [35]. The HA-BICD2 plasmid was a gift from Dr. C. Hoogenraad (Utrecht University, Utrecht, Netherlands). Site directed mutagenesis was used to create the siRNA-resistant HA-BICD2 and HA-A/V BICD2. The HA-mCherry and the HA-(Δ1–67) L2-3xFLAG constructs were generated in the study by Harwood et al., 2023 [24]. BICD2 was subcloned from the HA-BICD2 vector and inserted into either the HA-mCherry vector or the pCMV-FLAG vector. pcDNA3.1(-) EGFP-FLAG was described as previously [53].

## HPV pseudovirus production

HPV16.L2F, HPV18.L2F, HPV5.L2F pseudoviruses (PsVs) were produced by using polyethyleneimine (PEI) (Polysciences Inc.) to co-transfect HEK 293TT with p16sheLL, p18sheLL, or p5sheLL and the indicated reporter construct (pcDNA3.1 expressing GFP with a C-terminal S-tag or phGluc expressing GFP and secreted *Gaussia* luciferase). For the experiment shown in S4 Fig, EdU (5-ethynyl 2´-deoxyuridine) (Invitrogen, C10337) was used to label the reporter plasmid packaged in PsV. To produce EdU-labelled HPV16 PsV, EdU was added at 100 µM at 24 hours post-transfection of 293TT cells. Packaged pseudoviruses were purified by density gradient centrifugation in OptiPrep (Millipore Sigma) as described previously [33,34]. The purity of the pseudovirus preparations was analyzed by SDS-PAGE and Coomassie staining for L1 and L2.

## Immunoprecipitation and mass spectrometry

The IP-MS results shown in Fig 1A are from Harwood et al., 2023 [24]. See the Materials and Methods section for a detailed protocol.

## siRNA transfection

siRNAs used in this study were purchased from Dharmacon or Sigma. The target sequences and concentrations used are listed in Table 2. AllStar Negative (Qiagen) was used as a scrambled control siRNA. All cells were seeded and simultaneously reverse transfected using Lipofectamine RNAi MAX (Thermo Fisher Scientific). KD studies were carried out for 24 or 48 hours, as indicated.

## DNA transfection

For KD-rescue experiments, 1 x $10^5$ HeLa cells per well (in a 6-well plate) were transfected for 24 hours using the FuGENE HD (Promega) transfection reagent. 0.25–1 µg of plasmid expressing the indicated construct was used. For overexpression IP experiments, HeLa cells were seeded in 10-cm plates and allowed to reach ~70% confluency before transfection for 24 hours using FuGENE HD. Plasmids expressing indicated constructs were transfected at 3 µg.

**Table 2. siRNAs used for knockdowns.**

| siRNA | Sequence (5'-3') | Conc. |
|---|---|---|
| BICD1 siRNA #1 | CAUCGAAGGAGGCUUACUA | 50nM |
| BICD1 siRNA #2 | UCACUAAUGUACAGGCAGA | 50nM |
| BICD2 siRNA #1 | AGACGGAGCUGAAGCAGUU | 100nM |
| BICD2 siRNA #2 | UGAUGAAGCUGCGCAAUGA | 10-50nM |
| BICDR1 siRNA #1 | GCACUUAGAGCAAGAGAAA | 100nM |
| BICDR1 siRNA #2 | AUAAGGAGCUGACAGACAA | 100nM |
| BICDR2 siRNA #1 | GCCGGUGGCUUUCUCAGUA | 100nM |
| BICDR2 siRNA #2 | GAUGAGAUCUCGCUGCAGC | 100nM |

For overexpression biotin peptide pulldown experiments, HeLa cells were seeded in 6-cm plates and allowed to reach ~70% confluency before transfection with 2 μg of expressing plasmids for 24 hours using FuGENE HD. For protein purification, HEK 293T cells were seeded in 10cm plates and allowed to reach ~70% confluency before transfection with 3 μg of expressing plasmids for 48 hours using PEI transfection reagent.

## Immunoprecipitation

In all IP studies with non-transfected cells, HeLa cells (or HeLa S3 where indicated) were grown to ~80% confluency and then infected with HPV16.L2F. In all IP studies with transfected cells, HeLa cells were grown to ~70% confluency before transfection and infection. At the indicated time points, the cells were washed three times with phosphate-buffered saline [pH7.4] (PBS). For time course experiments, the second PBS wash contained 300 mM NaCl to remove extracellular HPV before cell lysis. For the dynein IP experiments, cells were lysed in 1% Decyl Maltose Neopenyl Glycol (DMNG) (Anatrace) in HN buffer (50 mM Hepes and 150 mM NaCl) containing 1 mM PMSF and were incubated on ice for 10 min. Lysed cells were then centrifuged for 10 min at 16,100g and the resulting supernatant was incubated with dynein IC antibody or an equal concentration of mouse IgG control antibody overnight at 4˚C. Pierce protein A/G agarose beads (Thermo Fisher Scientific) were then added to the samples for 30 min at 4˚C, followed by four washes with 0.1% DMNG in HN buffer and incubation at 95˚C for 10 min in 5x SDS sample buffer with 2-mercaptoethanol. In 3F Fig, cells were seeded and reverse transfected with the indicated siRNA for 48 hours before treatment as described above. For the BICD2 and HA IP experiments, cells were lysed in RIPA buffer (50 mM Tris pH 7.4, 150 mM NaCl, 0.25% sodium deoxycholate, 1% NP40, and 1 mM EDTA) containing 1 mM PMSF. Cells were incubated on ice for 10 min followed by centrifugation at 16,100 g for 10 min. Supernatants were incubated with BICD2, HA, or the matching control IgG antibody overnight at 4˚C. In the BICD2 IP experiments, samples were then treated as described above before three washes in RIPA buffer. In the HA IP experiments, Protein G magnetic Dynabeads (Thermo Fisher Scientific) were then added to the samples for 30 min at 4˚C and washed three times with RIPA buffer before elution as described above. In all experiments, cells were seeded in 10 cm plates and lysed in 400 μL of indicated lysis buffer. After centrifugation, 10% of the supernatant was taken for input and the remaining supernatant was incubated with the described antibodies.

## Proximity ligation assay

$0.35 \times 10^5$ HeLa S3 cells per well were plated in 24-well plates containing glass coverslips 48 h prior to infection. Approximately 6 h later, cells were transfected with 6.7 nM of indicated

siRNA (Table 2) as described above except using Lipofectamine RNAiMAX (Invitrogen). At 40–48 h after transfection, cells were infected with PsVs at MOI of ~200 in unmodified HeLa cells. For XXI treatment experiments, $0.6 \times 10^5$ HeLa S3 cells per well were plated in 24-well plates containing glass coverslips 20 h prior to infection. DMSO (0.2% as final concentration) or 2 µM XXI dissolved in DMSO were added to the medium 30 min prior to infection. At indicated times post-infection, cells were fixed with 4% paraformaldehyde (Electron Microscopy Sciences) at room temperature (RT) for 12 min, permeabilized with 1% Saponin (Sigma-Aldrich) at RT for 35–40 min, and blocked using DMEM10 at RT for 1–1.5 h. Cells were then incubated overnight at 4˚C with a pair of mouse and rabbit antibodies: a mouse antibody recognizing L1 (BD Biosciences, 554171, 1:1,000 when used with anti-TGN46, 1:500 when used with anti-dynein, 1:200 when used with other antibodies) and a rabbit antibody recognizing a cellular protein (anti-TGN46, Abcam, ab50595, 1:600; anti-EEA1, Cell Signaling Technology, 2411, 1:75; anti-BicD2, Abcam, ab117818, 1:200; anti-dynein, Invitrogen, PA5-89505, 1:200). PLA was performed with Duolink reagents (Sigma Aldrich) according to the manufacturer's instructions. Briefly, cells were incubated in a humidified chamber at 37˚C with a pair of PLA antibody (mouse and rabbit) probes for 75 min, with ligation mixture for 45 min, and then with amplification mixture for 3 h, followed by series of washes. Nuclei were stained with 4,6-diamidino-2-phenylindole (DAPI). Cellular fluorescence was imaged using the Zeiss LSM980 confocal microscope. Images were processed using a Zeiss Zen software version 3.1 and quantified using Image J Fiji version 2.3.0/1.53f.

## Protein purification

HEK 293T cells were seeded in a 10-cm plate and transfected with plasmids expressing EGFP-FLAG, FLAG-BICD2, HA-(Δ1–67) L2-3xFLAG or, in the case of retromer, VPS29-FLAG, VPS26-Myc and VPS35-Myc together. After 48 hours, cells were washed three times with PBS, harvested, and lysed in 1% Triton X-100 in HN buffer with 1 mM PMSF. Cells were incubated on ice for 10 min then centrifuged for 10 min at 16,100 g. The resulting supernatant was incubated with anti-FLAG M2 agarose beads (Millipore Sigma) for two hours at 4˚C. The beads were then washed twice with 1% Triton X-100 in HN buffer supplemented with 1 mM PMSF and 1 mM ATP to remove contaminating proteins, then once with 0.1% Triton X-100 in HN buffer supplemented with 1 mM ATP. The proteins were then eluted twice with 3xFLAG peptide in 0.1% Triton in HN buffer at room temperature for 30 min. The quality and quantity of the purified proteins were analyzed by SDS-PAGE and SimplyBlue SafeStain (Invitrogen) alongside BSA concentration standards.

## *In vitro* binding

~200 ng of purified FLAG-BICD2 was incubated with ~80 ng of purified HA-(Δ1–67) L2-3xFLAG or ~800 ng of EGFP-FLAG in a buffer composed of 0.1% Triton and 1mM PMSF in HN buffer for one hour at 37˚C with mild agitation. BICD2 antibody was added, and the samples incubated for an additional 30 min at 37˚C with mild agitation. The samples were then added to protein G magnetic Dynabeads and incubated for 20 min at 37˚C with mild agitation. The beads were washed with the reaction buffer and incubated in SDS sample buffer with 2-mercaptoethanol for 10 min at 95˚C, followed by SDS-PAGE and immunoblotting.

## HPV infection

Infections were performed asynchronously. For flow cytometry experiments, HeLa or HaCaT cells were treated as indicated and infected at a MOI ~ 1 with HPV16.L2F containing a GFP reporter plasmid or HPV18.L2F or HPV5.L2F containing a reporter plasmid expressing both

GFP and luciferase. 48 hours after infection, cells were washed with PBS, trypsinized, and resuspended in PBS with 2% FBS and 0.1 µg/mL DAPI. The Bio-Rad ZE5 cell analyzer was used to perform flow cytometry (University of Michigan Flow Cytometry Core Facility). The population of single, DAPI-negative cells were analyzed for GFP fluorescence. To analyze HPV infection by microscopy in KD-rescue experiments, HeLa cells were treated as indicated and infected at an MOI ~ 1.5 with HPV16.L2F containing a GFP reporter plasmid. 48 hpi, cells were washed with PBS, and stained for HA, as described below. GFP was visualized without antibody. For each experiment, the number of GFP-positive cells in 100 HA-positive cells were counted using the Nikon Eclipse TE2000-E inverted epifluorescence microscope.

## Biotin peptide pulldown

The peptides were purchased from GenScript or Promega and dissolved in DMSO. Stocks were diluted to 5 mg/mL and stored at -80˚C. For the biotin peptide pulldown experiments using purified proteins, ~100 ng of FLAG-BICD2, ~400 ng of EGFP-FLAG, or ~100 ng (S1E Fig)—~700 ng (S1D Fig) of the retromer complex were incubated with 10 µg of biotinylated peptide in Hepes buffer (1% Triton X-100, 20 mM Hepes pH 7, 50 mM NaCl, 5 mM MgCl$_2$, and 1 mM DTT) containing protease inhibitor cocktail (Thermo Fisher Scientific) for two hours at 4˚C. 30 µL of Pierce streptavidin magnetic beads (Thermo Fisher Scientific) were added to the samples and incubated for an additional hour at 4˚C. The beads were then washed three times with Hepes buffer and incubated at 95˚C for 10 min in SDS sample buffer with 2-mercaptoethanol. For experiments using whole cell extract instead of purified proteins, uninfected HeLa cells in a 6-cm dish were washed with PBS and lysed with 165 µL of Hepes buffer for 45 min on ice followed by centrifugation at 14,000g for 10 min. The resulting supernatant was incubated with peptide as described above and captured with 50 µL of streptavidin magnetic beads.

## Immunofluorescence

HeLa cells were seeded onto glass coverslips in a 6-well plate and treated as indicated. Cells were washed three times with PBS, fixed in 4% paraformaldehyde for 20 min at room temperature then washed four times with PBS. Permeabilization was carried out for 20 min at room temperature with tris-buffered saline (TBS)/0.2% Triton X-100/3% BSA followed by three washes with TBS-T (TBS with 0.1% Tween-20). The cells were blocked with TBS (with 0.2% Tween-20 and 3% BSA) for one hour at room temperature. Primary antibodies were diluted in TBS (with 0.2% Tween-20 and 3% BSA) and incubated with the coverslips overnight at 4˚C. For co-localization studies, samples were prepared in the same way as described in the section describing Proximity ligation assay. Primary antibodies recognizing EEA1 (Cell Signaling Technology, 2411) or TGN46 (Abcam, ab50595) were diluted with DMEM10 (1:250) and incubated with the coverslips overnight at 4˚C.Secondary antibodies were diluted with TBS containing 0.2% Tween-20 and 3% BSA or DMEM10 and incubated with coverslips for one hour at room temperature.

For cells infected with EdU-labeled PsV, cells were washed twice with PBS followed by a single wash with PBS [pH 10.7], then washed three times with PBS. EdU was detected by click chemistry according to the manufacturer's instructions (Invitrogen, C10337). Briefly, cells were washed twice with PBS containing 3% BSA, then permeabilized for 35 min at room temperature with PBS containing 1% saponin, followed by two washes with PBS containing 3% BSA. Then, cells were incubated with Click-iT reaction mixture for 30 min at room temperature, followed by a wash with PBS containing 3% BSA. Cells were blocked using DMEM10 at RT for 1–1.5 h and were then incubated overnight at 4˚C with an antibody recognizing

TGN46. Cells were incubated for one hour with secondary antibodies diluted in DMEM10. Coverslips were mounted with mounting medium containing DAPI (Abcam, ab104139). Images were taken with confocal microscopy (Zeiss LSM 800 or 980 confocal laser scanning microscope with a Plan-Apochromat 40x/1.4 oil differential interference contrast M27 objective). Representative images were chosen out of three independent experiments.

## Cell membrane integrity

Cell membrane integrity was assessed as described in Harwood et al., 2023 [24]. Briefly, non-permeabilized cells were washed with PBS, trypsinized, and resuspended in PBS supplemented with 2% FBS and 0.1 µg/ml DAPI. Flow cytometry was performed on the Bio-Rad ZE5 cell analyzer (University of Michigan Flow Cytometry Core Facility) to assess the percentage of DAPI-negative single cells.

## Cell cycle analysis

The cell cycle was analyzed as described in Harwood et al., 2023 [24]. Briefly, nonpermeabilized cells were washed with PBS, trypsinized, and resuspended in PBS supplemented with 2% FBS and 0.1 µg /ml DAPI. Flow cytometry was performed on the Bio-Rad ZE5 cell analyzer (University of Michigan Flow Cytometry Core Facility) to analyze DNA content using Hoechst 33342. FlowJo software (BD) was used to determine the percentage of cells in $G_1$, S, or $G_2$-M in control cells and gating was applied to treated cells.

## Statistical analysis

Statistical analyses represent data combined from at least three independent experiments. Data are presented as the mean values, with error bars representing standard deviation (SD). For infection and cell membrane integrity studies, results were analyzed with a two-tailed, unequal variance $t$ test. For cell cycle analysis studies, results were analyzed with a two-tailed, paired $t$ test. $p < 0.05$ was considered to be significant.

## Supporting information

**S1 Fig. A C-terminal amino acid region of L2 binds directly to BICD2 and retromer (related to Figs 2 and 3).** A. Whole cell extracts derived from HeLa cells were incubated with biotinylated L2 peptide C or M. The peptides were precipitated by streptavidin beads and the precipitated material was subjected to SDS-PAGE and immunoblotting with an antibody recognizing BICD2 to detect endogenous BICD2. B. As (A) except using whole cell extracts derived from HeLa cells transfected with HA-BICD2-mCherry or the control HA-mCherry with biotinylated L2 peptide C or C442, or no peptide as a control. C. Coomassie stain of the retromer complex (VPS29-FLAG, VPS35-Myc, and VPS26-Myc). D. As (A) except using purified retromer with the indicated biotinylated L2 peptide (please see peptides in Fig 2C). The precipitated material was subjected to SDS-PAGE and immunoblotted with an antibody recognizing VPS26. All samples were electrophoresed on the same gel. Irrelevant lanes were removed. E. Biotinylated peptide C was incubated with the retromer complex in the presence or absence of FLAG-BICD2. The peptide was precipitated as described above and immunoblotting with an antibody recognizing FLAG was performed. VPS29-FLAG indicates the presence of the retromer complex in the pellet. F. As in (A) except using only biotinylated L2 peptide C or no peptide as a control and immunoblotting for endogenous BICDR1. (TIF)

**S2 Fig. Effects of BICD2 knockdown during HPV infection (related to Fig 3). A.** Whole cell extracts derived from HeLa cells transfected with indicated siRNA were subjected to SDS-PAGE and immunoblotting with the indicated antibodies. Actin, loading control. **B.** HeLa cells transfected with the indicated siRNA were trypsinized and incubated in a buffer containing DAPI. The fraction of DAPI-negative cells was measured by flow cytometry. The results were normalized to the fraction of untreated, DAPI-negative cells. $^*p \leq 0.05$ **C.** As in (A), except using HaCaT cells. **D.** As described in (B), except using HaCaT cells. **E.** HeLa cells transfected with the indicated siRNA were fixed, permeabilized, and subjected to immunofluorescent staining with an antibody recognizing GM130 (green). Nuclei were stained with DAPI (blue). Representative images taken by confocal microscopy are shown. **F.** HeLa S3 cells were transfected with scrambled control (siScram) or BICD2-targeting (siBICD2) siRNAs. Cells were stained with antibodies recognizing EEA1 and TGN46. Immunofluorescence images were shown; EEA1, green; TGN46, magenta; nuclei (DAPI), blue. Pearson's correlation coefficient values for EEA1 and TGN46 colocalization in those cells are shown. Each dot represents an individual cell (n>30) and black horizontal lines indicate the mean value of the analyzed population in each group. ns, not. significant. The graph shows results of a representative experiment. Similar results were obtained in two independent experiments. **G.** HeLa cells transfected with the indicated siRNA were stained for cellular DNA by incubation with Hoechst 33342 then trypsinized and analyzed by flow cytometry for relative Hoechst 33342 fluorescence. One set of representative histograms is shown from three independent experiments, with the percentage of cells in $G_1$, S, or $G_2$-M phases indicated. **H.** Means of cell cycle phase distributions as in (F) from three independent experiments. N.S., not significant. BICD2 siRNA #2 was used in panels E-F.
(TIF)

**S3 Fig. The C-terminus segments of L2 from different types of HPV bind BICD2 (related to Fig 3). A.** L2 amino acids of HPV5 and HPV18 were aligned with HPV16 L2 C-terminus segment 434–473. The asterisk indicates identity, colon indicates conservative substitution, and period indicates semi-conservative substitution. **B.** Sequences of the L2 peptides of HPV16, HPV5 and HPV18. B indicates biotin. In HPV16 peptide C, amino acids for the retromer binding site are shown in green and those for the CPP are shown in blue. **C.** FLAG-BICD2 was incubated with the indicated biotinylated L2 peptides. Precipitation was performed using streptavidin beads. The precipitated material was subjected to SDS-PAGE and immunoblotted with an antibody recognizing FLAG.
(TIF)

**S4 Fig. BICD2 depletion results in encapsidated DNA accumulation in the TGN/Golgi (related to Fig 5).** HeLa S3 cells were transfected with siScram or siBICD2 siRNA and infected with EdU-labeled HPV at the MOI of ~100. At 24 hpi, cells were stained with antibodies recognizing TGN46, and EdU was detected as described in Materials and Methods section. Immunofluorescence images were shown; EdU, green; TGN46, magenta; nuclei (DAPI), blue. Pearson's correlation coefficient values for EdU and TGN46 colocalization in those cells are shown. Each dot represents an individual cell (n>30) and black horizontal lines indicate the mean value of the analyzed population in each group. $^{***}p < 0.001$. The graph shows results of a representative experiment. Similar results were obtained in two independent experiments. BICD2 siRNA #2 was used.
(TIF)

## Acknowledgments

We would like to thank Yuka Takeo for providing purified retromer.

## Author Contributions

**Conceptualization:** Kaitlyn Speckhart, Jeongjoon Choi, Daniel DiMaio, Billy Tsai.

**Data curation:** Kaitlyn Speckhart, Jeongjoon Choi.

**Formal analysis:** Kaitlyn Speckhart, Jeongjoon Choi.

**Funding acquisition:** Kaitlyn Speckhart, Daniel DiMaio, Billy Tsai.

**Investigation:** Kaitlyn Speckhart, Jeongjoon Choi.

**Methodology:** Kaitlyn Speckhart, Jeongjoon Choi, Daniel DiMaio, Billy Tsai.

**Project administration:** Kaitlyn Speckhart, Jeongjoon Choi, Daniel DiMaio, Billy Tsai.

**Resources:** Billy Tsai.

**Supervision:** Daniel DiMaio, Billy Tsai.

**Validation:** Kaitlyn Speckhart, Jeongjoon Choi.

**Visualization:** Kaitlyn Speckhart, Jeongjoon Choi, Billy Tsai.

**Writing – original draft:** Kaitlyn Speckhart, Jeongjoon Choi, Daniel DiMaio, Billy Tsai.

**Writing – review & editing:** Kaitlyn Speckhart, Jeongjoon Choi, Daniel DiMaio, Billy Tsai.

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
