## [Decision Letter · Decision Letter 0]

18 Dec 2023

Dear Prof. Tsai,

Thank you very much for submitting your manuscript "The BICD2 dynein cargo adaptor binds to the HPV16 L2 capsid protein and promotes HPV infection" for consideration at PLOS Pathogens. As with all papers reviewed by the journal, your manuscript was reviewed by members of the editorial board and by several independent reviewers. In light of the reviews (below this email), we would like to invite the resubmission of a significantly-revised version that takes into account the reviewers' comments.

We cannot make any decision about publication until we have seen the revised manuscript and your response to the reviewers' comments. Your revised manuscript is also likely to be sent to reviewers for further evaluation.

Sincerely,

Richard B.S. Roden

Academic Editor

PLOS Pathogens

Alison McBride

Section Editor

PLOS Pathogens

Kasturi Haldar

Editor-in-Chief

PLOS Pathogens

orcid.org/0000-0001-5065-158X

Michael Malim

Editor-in-Chief

PLOS Pathogens

orcid.org/0000-0002-7699-2064

Reviewer's Responses to Questions

**Part I - Summary**

Reviewer #1: Intracellular trafficking of HPV is an important area of study, in which many details are still lacking. Speckhart et al. report a direct interaction between the papillomavirus L2 protein C-terminus and the dynein motor adapter BICD2, and investigate the role of BICD2 in HPV infection. Following up on prior published IP-MS/MS data wherein they identified the dynein motor heavy chain as an interaction partner with the L2 minor capsid protein, they show that the dynein adaptor BICD2 coIPs with L2 in a membrane spanning manner as early as 8h post infection. This interaction is also detected by PLA in fixed cells. They show direct binding between L2 and BICD2 using purified proteins and use biotinylated peptides to show that binding can occur via a 20-residue fragment very close to the C-terminus of L2. siRNA silencing and complementation experiments (including a dynein binding-defective BICD2 mutant) support a role for L2-BICD2-dynein interaction in promoting HPV infection. PLA experiments suggest that upon siRNA depletion of BICD2, HPV accumulates at both endosomes (EEA1+L1 PLA) and Golgi (TGN46 +L1 PLA) at 16h, 24h, and 32h post infection.

Overall the paper is well written and has a logical flow. They present solid data supporting a direct interaction between a specific C-terminal region of L2 and the BICD2 dynein adapter, suggesting that BICD2 and the L2-BICD2-dynein interaction supports intracellular trafficking of HPV, specifically the post-endosome and post-Golgi trafficking. This work is in agreement with the prior identification of similar L2-dynein interactions (ref 24) and further advances this field with new knowledge of the BICD2 adaptor. The work could be strengthened by further mapping the L2-BICD2 interaction(s), further dissecting the trafficking defect through additional experiments and controls, and identification of an L2 mutant that can span, interact with retromer, but is defective in binding BICD2 (unclear if this was attempted).

Reviewer #2: The authors provided convincing data to show that the dynein cargo adaptor BICD2 binds directly to the HPV L2 capsid protein during entry, recruiting HPV to dynein and transporting the virus along the endosome�TGN/Golgi axis to promote infection. They use SiRNA and mutation strategies to show that in the absence of BICD2 function, HPV accumulates in the endosome and TGN and infection is inhibited. They also identified a short segment near the C-terminus of L2 (aa 442-461) is sufficient for direct binding with BICD2. The study is well presented and the manuscript is well-written.

Reviewer #3: The manuscript ‚The BICD2 dynein cargo adaptor binds to L2 capsid protein and promotes GPV infection’ by Speckhardt and co-workers investigates the role of dynein and the BICD2 adaptor in the entry of human papillomavirus type 16 (HPV16). HPV16 enters cells by endocytosis, is routed through the endosomal system to the trans-Golgi network (TGN), and accesses the nuclear lumen during mitosis after nuclear envelope breakdown. The premise of this study is that HPV16 like all viruses is devoid of locomotive capability, so that the retrograde trafficking of the virus must be assisted by the cellular trafficking machinery.

Almost all viruses use microtubular transport and its retrograde motor dynein for this. Indeed, quite a few previous studies have already shown that dynein facilitates the transport of HPVs. Here, the authors confirm these findings and show that L2 can interact with dynein directly. Moreover, using biochemical assays, loss-of-function infection assays, and immunocytochemistry/microscopy employing proximity-ligation assays the authors implicate cellular BICD2 as dynein adaptor linking L2 and dynein for intracellular transport.The data itself appears sound, and is overall well executed. However, while the findings are supported by the data presented in the manuscript and interesting for the papillomavirus community, they appear to be only incremental to our existing understanding of HPV entry, in particular, since the data fails to provide sufficient evidence for a convincing model on intracellular trafficking of HPVs including and reconciling it with previous work.

**Part II – Major Issues: Key Experiments Required for Acceptance**

Reviewer #1: The peptide binding data convincingly shows that BICD2 can bind to a 20-mer peptide from L2. A missed opportunity was to further dissect this binding and define the specific residues of L2 that necessary, with the hope of identifying a mutant that is specifically defective for binding to BICD2 but otherwise functional for retromer binding and membrane spanning. The conclusion that this small region is sufficient for dynein/BICD2 binding would be solidified if the in vitro binding assays from Figure 2 were repeated with mutation of the larger recombinant L2 (delta1-67) fragment, and with mutant virions during infection. Do peptides #3 and #4 still bind BICD2 if the native C-terminal sequences are present? Can these peptides simultaneously bind retromer and BICD2/dynein?

The conclusion that HPV trafficking is blocked at both the endosome and Golgi is confusing and only relies on PLA data. An important control that is missing is to determine if BICD2 silencing alters the distribution of either EEA1 or TGN46. With regards to TGN46, this is a transmembrane protein localized to Golgi at steady state but can be secreted to the plasma membrane and re-enter via endosomes before retrieval back to the Golgi, quite different than the GM130 Golgi matrix protein that doesn’t appear to change distribution upon BICD2 knockdown (Fig S2E). If BICD2 silencing causes more TGN46 protein to be within endosomes (increases TGN46 + EEA1 PLA and/or localizes TGN46 to endosomes by conventional IF microscopy) then that could greatly affect the interpretation of the PLA data. Alternate methods to look at trafficking may also strengthen the paper.

Ciliobrevin D treatment should perturb mitosis, which is needed for post-Golgi trafficking and infection. Proper cell cycle analysis and a growth curve in the presence of drug is needed to rule out this possibility. Another dynein inhibitor called dynarrestin was reported to induce spindle misorientation and pseudoprometaphase delay (PMID 29396292), so these drugs should be used cautiously, or limited to studies on early pre-mitotic trafficking. On the topic of cell cycle, the data in S2F show a difference in G2/M/G1 ratios upon BICD2 knockdown, although these did not reach statistical significance. The histograms shown in S2F have a different shape, and there is an additional shoulder of cells with >2n DNA content upon silencing of BICD2. I suggest a more rigorous analysis to ensure these cells are cycling properly through mitosis, with normal kinetics. Perhaps a cellular growth curve comparing siScram to siBICD2, and giving the siRNA more time to work before doing the cell cycle analysis. Dynein plays a role in mitosis and it seems that silencing an adaptor like BICD2 would perturb the kinetics of mitosis.

Reviewer #2: 1) The authors identified a 19-amino acid (442-461) C-terminal segment of HPV16 L2 that is sufficient for binding to BICD2. How conserved is the BICD2 binding region of L2 among HPV types? It would be helpful to show the alignment of this region among different HPV types. It would be interesting to know which 19 amino acid (s) are key binding epitopes for BICD2.

2) It would be interesting to know how HPV L2 without BICD2 binding epitopes infects and transports inside cells.

Reviewer #3: A major concern is that the presented model remains for a good part speculative and unsubstantiated. The authors propose that BICD2/dynein already engage L2 upon membrane penetration in endosomes to facilitate transport to the TGN, and BICD2/dynein would promote budding of vesicles from the TGN at mitosis onset to facilitate nuclear import. While the first part is partially supported by the data showing that BICD2 knockdown reduces in part trafficking to the TGN, it remains unclear whether other adaptor proteins assist or compensate this transport step. For the second part, the authors do not provide any evidence for a putative budding step. While BICD2 knockdown accumulates virus in the Golgi, and this may be caused by failure to bud, it would also be possible that the BICD2 and dynein simply prevent dissemination of the virus from a secretory organelle such as the Golgi, thereby keeping it in place to facilitate nuclear entry upon mitosis onset. Potentially, BICD2/dynein may also assist onward trafficking during mitosis, but published work already implicated RanBP10/KPNA2 as adaptors for a dynein mediated transport step at this juncture, and it would be of great interest in addressing how the two adaptors work together or in an alternating fashion. What would cause L2 to switch between adaptors? Also, since the putative interaction site appears to be engaged by several different cellular proteins, I wonder how these interactions would be orchestrated – as I find it very difficult to envision simultaneous interactions at sites that are spatially not at all or not well separated. From a virological point of view, it would be of interest to address whether several distinct HPV types would engage the same adaptor(s) for transport. Addressing these (mechanistic) questions experimentally would greatly improve the manuscript would be of interest to a broader readership.

**Part III – Minor Issues: Editorial and Data Presentation Modifications**

Reviewer #1: Peptide #4 in Fig 2 is 20 residues, not 19.

Reviewer #2: 3) The authors used ciliobrevin D (cilio D) to inhibit the ATPase activity of dynein and showed that binding with BICD2 is critical for infection of multiple HPV types. A novel inhibitor Dynarrestin with less toxicity was reported and is worth testing in your system ( https://www.cell.com/cell-chemical-biology/pdf/S2451-9456(17)30463-4.pdf )

Reviewer #3: Technically, most of the assays are sound. Given that the proximity ligation assay is the sole line of evidence for transport failures, a second line of investigation such as following the viral DNA by immunofluorescence and colocalization analysis may improve clarity on how severe the phenotypes upon perturbation are.

The referencing looks in part a bit random. While in part numerous publications supporting a statement are referenced, in other places only few and sometimes not the first observations are cited.

PLOS authors have the option to publish the peer review history of their article (what does this mean?). If published, this will include your full peer review and any attached files.

Reviewer #1: No

Reviewer #2: No

Reviewer #3: No

Figure Files:

Data Requirements:

Reproducibility:

To enhance the reproducibility of your results, we recommend that you deposit your laboratory protocols in protocols.io, where a protocol can be assigned i

---

## [Decision Letter · Decision Letter 1]

22 Apr 2024

Dear Prof. Tsai,

Thank you very much for submitting your manuscript "The BICD2 dynein cargo adaptor binds to the HPV16 L2 capsid protein and promotes HPV infection" for consideration at PLOS Pathogens. As with all papers reviewed by the journal, your manuscript was reviewed by members of the editorial board and by several independent reviewers. The reviewers appreciated the attention to an important topic. Based on the reviews, we are likely to accept this manuscript for publication, providing that you modify the manuscript according to the review recommendations.

Reviewer 3 has lingering concerns about the proposed model that should be clarified and addressed in the discussion. Appropriate caveats should be added.

Sincerely,

Richard B.S. Roden

Academic Editor

PLOS Pathogens

Alison McBride

Section Editor

PLOS Pathogens

Michael Malim

Editor-in-Chief

PLOS Pathogens

orcid.org/0000-0002-7699-2064

Reviewer 3 has lingering concerns about the proposed model that should be clarified and addressed in the discussion. Appropriate caveats should be added.

Reviewer Comments (if any, and for reference):

Reviewer's Responses to Questions

**Part I - Summary**

Reviewer #1: The revised manuscript by Speckhart et al. has been greatly improved and all prior concerns have been well addressed. The conclusions are well supported by rigorous experimental data and the discussion is balanced without too much speculation.

Reviewer #2: The revised version of “The BICD2 dynein cargo adaptor binds to the HPV16 L2 capsid protein and promotes HPV infection” has added additional data to support their previous findings. For example, they assessed trafficking using PsV containing EdU-labeled DNA to show a similar result to the PLA data. It also clarified some confusions on a putative budding step under BICD2 KD and provided explanations in the discussion section on potential mechanisms of how (or whether) these proteins work together to support infection.

Reviewer #3: The revised manuscript ‚The BICD2 dynein cargo adaptor binds to L2 capsid protein and promotes GPV infection’ by Speckhardt and co-workers investigates the role of dynein and the BICD2 adaptor in the entry of human papillomavirus type 16 (HPV16).

My major concern was that the presented model is not well substantiated and related to the existing literature. While the data supports the potential of BICD2 and now also BICDR2 to bind to L2 to the retromer and CCP peptide, and BICD2 knockdown leads to a transient accumulation of L1 in early endosomes, and a more pronounced accumulation in the trans-Golgi network, but how it does so remains somewhat speculative. The authors have clarified my concern on the budding part, but I have still considerable doubts on their model (see below).

**Part II – Major Issues: Key Experiments Required for Acceptance**

Reviewer #1: none

Reviewer #2: None

Reviewer #3: 1. If BICD2 binds to penetrated L2 and is responsible for transport, why are the effects on endosomal retention so small particularly, if it would associate already in early endosomes? This means, that there are at least additional adaptors facilitating routing of the virus. Why didn’t the authors attempt to do a double knockdown of BICD2 and BICDR2, if those are considered to be the main dynein adaptors at this point? Also the new data on EdU labelled viral DNA (Suppl. Fig.2) does not support any early endosome delay.

2. Alternatively or additionally, the effects of BICD2 knockdown may account for general transport defects along the endosomal system or within endosome to Golgi transport. It is unfortunate that the epitopes of the interacting peptide cannot be resolved from residues crucial for penetration and retromer binding, but this would have resolved the question.

3. Relying entirely on PLA stainings was another critical point. The authors now provide images following EdU labelled viral DNA. To my surprise (given the considerable signal for EdU), there is no signal in the nucleus and little overlap with the TGN for the scrambled siRNA, and inly incremental increase in BICD2 depleted cells? This inspires little confidence that their model would be correct. One would have expected a notable drop in intranuclear localisation. Where is the viral DNA at this point?

**Part III – Minor Issues: Editorial and Data Presentation Modifications**

Reviewer #1: none

Reviewer #2: Additional in vivo studies (e.g. the preclinical PV models) would further strengthen these in vitro findings.

Reviewer #3: I would highly recommend to discuss the existing literature on dynein-mediated transport and the possible scenarios/implications within one paragraph rather than at different places within the discussion. It is hard for a non-expert on this matter to follow otherwise.

PLOS authors have the option to publish the peer review history of their article (what does this mean?). If published, this will include your full peer review and any attached files.

Reviewer #1: No

Reviewer #2: No

Reviewer #3: No

Figure Files:

Data Requirements:

Reproducibility:

References:

---

## [Decision Letter · Decision Letter 2]

24 May 2024

Dear Prof. Tsai,

We are pleased to inform you that your manuscript 'The BICD2 dynein cargo adaptor binds to the HPV16 L2 capsid protein and promotes HPV infection' has been provisionally accepted for publication in PLOS Pathogens.

Best regards,

Richard B.S. Roden

Academic Editor

PLOS Pathogens

Alison McBride

Section Editor

PLOS Pathogens

Michael Malim

Editor-in-Chief

PLOS Pathogens

orcid.org/0000-0002-7699-2064

Reviewer Comments (if any, and for reference):

Reviewer's Responses to Questions

**Part I - Summary**

Reviewer #1: All prior concerns have been well addressed. The revised conclusions are supported by rigorous experimental data and the discussion is balanced and thorough.

Reviewer #2: I agree with the other two reviewers that the paper is well-written and logically structured. The data are well-executed overall. While the findings from the current study represent incremental advancements in our understanding of HPV entry, future studies could enhance this project. Suggestions include further mapping of the L2-BICD2 interactions, a more detailed dissection of the trafficking defect using additional experiments and controls, and the identification of an L2 mutant capable of spanning and interacting with retromer but defective in BICD2 binding.

Reviewer #3: Overall, the authors addressed most of my lingering concerns by more clearly stating specific uncertainties in the revised discussion. However, I still think that a lack of nuclear import would have been a great addition to the manuscript - it remains rather surprising to me that nuclear import is not observed at the given times of the assay in the wild type situation. While not unambiguously demonstrating their model, given the sum of data presented the authors provide a plausible working hypothesis.

**Part II – Major Issues: Key Experiments Required for Acceptance**

Reviewer #1: none

Reviewer #2: The authors provide additional interpretations of their data. They propose conducting experiments to study the interaction of two different cellular proteins. However, it would be a challenging experiment to envisioning simultaneous interactions at sites that are spatially not well separated. I agree that testing whether several distinct HPV types would engage the same adaptor(s) for transport would enhance the manuscript and be of interest to a broader readership. I hope they will continue to pursue this project.

Reviewer #3: (No Response)

**Part III – Minor Issues: Editorial and Data Presentation Modifications**

Reviewer #1: none

Reviewer #2: None

Reviewer #3: (No Response)

PLOS authors have the option to publish the peer review history of their article (what does this mean?). If published, this will include your full peer review and any attached files.

Reviewer #1: No

Reviewer #2: No

Reviewer #3: No

---

## [Editor Report · Acceptance letter]

29 May 2024

Dear Prof. Tsai,

We are delighted to inform you that your manuscript, "The BICD2 dynein cargo adaptor binds to the HPV16 L2 capsid protein and promotes HPV infection," has been formally accepted for publication in PLOS Pathogens.

Best regards,

Michael Malim

Editor-in-Chief

PLOS Pathogens

orcid.org/0000-0002-7699-2064